# Diterpenoids from *Plectranthus* spp. as Potential Chemotherapeutic Agents via Apoptosis

**DOI:** 10.3390/ph13060123

**Published:** 2020-06-16

**Authors:** Tomasz Śliwiński, Przemysław Sitarek, Ewa Skała, Vera M. S. Isca, Ewelina Synowiec, Tomasz Kowalczyk, Michał Bijak, Patrícia Rijo

**Affiliations:** 1Laboratory of Medical Genetics, University of Lodz, Pomorska 141/143, 90-236 Lodz, Poland; tomasz.sliwinski@biol.uni.lodz.pl (T.Ś.); ewelina.synowiec@biol.uni.lodz.pl (E.S.); 2Department of Biology and Pharmaceutical Botany, Medical University of Lodz, Muszynskiego 1, 90-151 Lodz, Poland; przemyslaw.sitarek@umed.lodz.pl (P.S.); ewa.skala@umed.lodz.pl (E.S.); 3Center for Research in Biosciences & Health Technologies (CBIOS), Universidade Lusófona de Humanidades e Tecnologias, 1749-024 Lisboa, Portugal; p5620@ulusofona.pt; 4Instituto de Investigação do Medicamento (iMed.ULisboa), Faculdade de Farmácia, Universidade de Lisboa, 1649-003 Lisboa, Portugal; 5Department of Molecular Biotechnology and Genetics, University of Lodz, Banacha 12/16, 90-237 Lodz, Poland; tomasz.kowalczyk@biol.uni.lodz.pl; 6Biohazard Prevention Centre, Faculty of Biology and Environmental Protection, University of Lodz, Pomorska 141/143, 90-236 Lodz, Poland; michal.bijak@biol.uni.lodz.pl

**Keywords:** *Plectranthus* sp., cytotoxic activity, apoptosis, DNA damage, mitochondrial membrane potential, ROS levels, gene expression

## Abstract

*Plectranthus* spp. is widely known for its medicinal properties and bioactive metabolites. The cytotoxic and genotoxic properties of the four known abietane diterpenoids: 7α-Acetoxy-6β-hydroxyroyleanone (Roy), 6,7-dehydroroyleanone (Deroy), 7β,6β-dihydroxyroyleanone6 (Diroy), and Parvifloron D (Parv), isolated from *P. madagascariensis* (Roy, DeRoy, and Diroy) and *P. ecklonii* (Parv) were evaluated. The tested compounds showed cytotoxic effects against the human leukemia cell line CCRF-CEM and the lung adenocarcinoma cell line A549. All tested compounds induced apoptosis by altering the level of pro- and anti-apoptotic genes. The results show that from the tested diterpenoids, Roy and Parv demonstrated the strongest activity in both human cancer cell lines, changing the permeability mitochondrial membrane potential and reactive oxygen species (ROS) levels, and possibly inducing mtDNA or nDNA damage. In conclusion, the abietane diterpenoids tested may be used in the future as potential natural chemotherapeutic agents

## 1. Introduction

Cancer is one of the most common causes of death worldwide [1,2]. The need for new and effective cytotoxic agents promotes continuous research on natural products. Many isolated plant compounds have been found to offer potential as treatments and may serve as alternatives to chemotherapy, which is associated with high costs and serious side effects [3,4,5,6]. These compounds might serve as novel drugs for use in combination therapy in various types of cancer [7]. The Lamiaceae family, namely the *Plectranthus* genus, is widely known for its medicinal properties due to the presence of secondary metabolites with an extensive and different range of biological activity [8,9]. Among many metabolites, diterpenes with an abietane skeleton attract special scientific interest as these are the most diverse type of diterpenoids isolated from the mentioned genus. Alongside this characteristic, many abietane diterpenoids possess cytotoxic properties. An example of such diterpenoids is parvifloron D (Parv) (Figure 1), which can be naturally found in *Plectranthus ecklonii* [10]. This abietane diterpene has already proven to be a strong cytotoxic plant-derived compound with broad activity, namely against melanoma, glioma, leukemia, pancreatic, breast, lung, colon, and P-glycoprotein-overexpressing cells [10,11,12,13,14,15,16,17].

7β,6β-Dihydroxyroyleanone (Diroy) and 7α-acetoxy-6β-hydroxyroyleanone (Roy) (Figure 1) are oxidized abietanes known for their antibacterial and cytotoxic potential and are commonly found in *P. madagascariensis* [12,18,19]. 6,7-Dehydroroyleanone (Deroy) (Figure 1) also demonstrated notable activity against human leukemia cells and glioma cells, although it does not have a natural occurrence in the previously mentioned plant species, but rather in its essential oil, in which it is the main component [12,18]. 

In this study, we present the cytotoxic and genotoxic properties of four abietane diterpenoids (Roy, Deroy, Diroy, and Parv) isolated from the *Plectranthus* genus in acute lymphocytic leukemia (CCRF-CEM) and lung adenocarcinoma cell lines (A549).

## 2. Results

### 2.1. 3-(4,5-Dimethylthiazol-2-yl)-2,5-diphenyl Tetrazolium Bromide (MTT) Assay

The present study tested the cytotoxic effect of four abietane diterpenoids isolated from *P.*
*madagacariensis* and *P. ecklonii*, on two cancer cell lines (CCRF-CEM and A549). Our results showed that for the CCRF-CEM cell line (Figure 2), a higher cytotoxic effect was observed for Roy with IC_50_ = 0.7 µg/mL. A lower cytotoxicity was revealed by Parv (IC_50_ = 1.15 µg/mL), Deroy (IC_50_ = 4.5 µg/mL), and Diroy (IC_50_ = 25 µg/mL). For the A549 cell line (Figure 3), the most cytotoxic effects were demonstrated by Parv with IC_50_ = 1.15 µg/mL. The other compounds, i.e., Roy, Deroy, and Diroy, revealed a less cytotoxic effect with IC_50_ 3, 6.25, and 25 µg/mL, respectively. Additionally, our results showed that Roy and Deroy had more anti-proliferative activity toward CCRF-CEM than A549. Furthermore, Diroy and Parv possessed similar cytotoxic activity for both tested cell lines. 

### 2.2. Reactive Oxygen Species (ROS) 

Reactive oxygen species level was measured in CCRF-CEM and A549 cells after 1, 2, 12, 24, and 48 h treatment with Roy, Deroy, Diroy, and Parv. After 12 h of A549 cell incubation with Deroy, the ROS level was higher than that of control cells, and this effect was maintained for up to 48 h (Figure 4). On the other hand, the A549 cells had a lower ROS level after treatment with Roy (Figure 4). For the CCRF-CEM cell line (Figure 5), a significant increase in ROS level was also observed after one hour of incubation with Deroy, and that level was maintained for two hours. It was also found that after 12 h of treatment with Diroy, CCRF-CEM cells had a higher ROS production, and this was still measurable for up to 48 h. Furthermore, after 48 h of incubation with Roy, an increase in the level of ROS was also observed. 

### 2.3. Mitochondrial Membrane Potential

Additionally, the mitochondrial membrane potential was checked after treatment with Roy, Deroy, Diroy, and Parv. In the A549 cell line, only Parv increased the mitochondrial membrane potential (Figure 6). In the CCRF-CEM cell line, all tested compounds decreased the mitochondrial membrane potential (Figure 7). 

### 2.4. Mitochondrial Copy Number

The CCRF-CEM cells demonstrated a higher mtDNA copy number than control cells after an hour of incubation with the tested compounds. The A549 cells also showed a slightly higher mtDNA. Data are shown in Figure 8.

### 2.5. Mitochondrial DNA Damage

The present study found an increase in lesion rate in the *ND5* region: 19.51 lesions per 10 kb DNA in A549 cells treated with Diroy for 24 h and 7.90 and 13.73 lesions per 10 kb DNA in CCRF-CEM cells treated with Deroy and Parv D for 24 h (Figure 9), respectively. Furthermore, a higher lesion rate was observed in the *ND1* region in CCRF-CEM cells after 24 h treatment with Diroy, Deroy, and Parv, resulting in 4.46, 8.86, and 7.35 lesions per 10 kb DNA, respectively. No difference was observed in the amount of mtDNA damage in the *ND1* region between samples and controls of A549 cells (Figure 9). 

### 2.6. Nuclear DNA Damage

In addition, we also investigated nDNA damage by semi-long run (SLR)-(quantitative RT) qRT-PCR amplification of DNA isolated from A549 and CCRF-CEM cells exposed to Roy, Deroy, Diroy, and Parv for 24 h. Our results (Figure 10) showed that in CCRF-CEM cells, Deroy increased nDNA damage quantities of 3.82 lesions/10 kb in the *HPRT1* region. DNA isolated from A549 cells treated with Roy and Diroy exhibited increased nDNA damage quantities of 4.78 and 4.76 lesions per 10 kb DNA in the *HPRT1* region, respectively. nDNA damage analysis of the *TP53* region revealed no DNA lesions in the samples treated with any tested compound. 

### 2.7. Gene Expression

Real-time PCR was used to determine the gene expression profile in A549 and CCRF-CEM cells. The study describes gene expressions (*Bax*, *Bcl-2*, *TP53,* and *Cas-3*) after 24 h of treatment with Roy, Deroy, Diroy, and Parv. Significant differences were found in the gene expression profiles between A549 or CCRF-CEM cells after treatment with compounds and untreated, control cells. Our results showed upregulation in the level of expression of *Bax, TP53,* and *Cas-3* for all tested compounds, and downregulation of *Bcl-2* on the A549 and CCRF-CEM cell lines (Figure 11A,B). 

## 3. Discussion

The plant kingdom is a rich source of active compounds with a broad spectrum of biological activity, and these are well represented in both modern and ancient pharmacotherapy. Even today, they are the subject of the ongoing search for new, more effective compounds of natural origin in the fight against illness, and particularly against cancer [20,21]. The WHO lists cancers such as leukemia and lung cancer as the second leading cause of death globally, accounting for an estimated 9.6 million deaths, i.e., one in six, in 2018 [22]. A wide range of active abietane diterpenes with various medical activities can be obtained from species of *Plectranthus* [13,14,15,16], a large and widespread genus with a diversity of ethnobotanical uses [23].

The abietane diterpenes Roy, Deroy, Diroy, and Parv were isolated from plants of the *Plectranthus* genus. The aim of the present study was to check whether these four abietane diterpenoids possess cytotoxic and genotoxic properties in a leukemia cell line (CCRF-CEM) and lung adenocarcinoma cell line (A549). The study evaluated the viability of cancer cells, mitochondrial membrane potential, ROS levels, mtDNA copy number, and DNA damages after treatment with these abietane diterpenoids. Our results demonstrate that all tested abietane diterpenoids Roy, Deroy, Diroy, and Parv showed cytotoxic activity in the tested range of concentrations (0–100 µg/mL) in the human leukemia cell line and lung adenocarcinoma. These diterpenoids showed concentration-dependent inhibition of cell viability in both tested cell lines. IC_50_ ranged from 0.7–25 µg/mL for the CCRF-CEM line to 1.15–25 µg/mL for the A549 line, indicating that the latter was more resistant to the tested compounds.

The strongest cytotoxic activity against the CCRF-CEM line was observed for Roy (IC_50_ = 0.7 µg/mL) and Parv (IC_50_ = 1.15 µg/mL). Parv and Roy showed similar cytotoxic effects against the lung cancer cell line (A549) with IC_50_ = 1.15 and 3 µg/mL, respectively. Our previous study showed a stronger cytotoxic effect for Parv (IC_50_ = 3.125 µg/mL) compared to Roy (IC_50_ = 25 µg/mL) against primary H7PX glioma cells [12]. A similar effect was shown by Saraiva et al. [16] for Parv for human breast cancer (MDA-MB-231) cells with IC_50_ = 2.48 µM. Additionally, Parv also demonstrated promising cytotoxic activity against human leukemia (HL-60, MOLT-3, K562, and U937) [10] and melanoma (human A375 and mouse B16VP) [11] cell lines. Burmistrova et al. [10] reported moderate cytotoxic properties of Roy against the MCF-7 human breast cancer line, with IC_50_ = 6.8 µM. Furthermore, Matias et al. [13] revealed that Roy had high selectivity for lung cancer cells with selectivity indexes of 3.2 µM. On the other hand, Roy did not display cytotoxic effects against other leukemia cell lines including HL-60, U937, and MOLT-3 [24], in contrast to the present study, and the human melanoma cell line SK-MEL1 was found to be resistant to Roy treatment [24]. On the other hand, our results indicate that the royleanone Deroy has moderate cytotoxic properties against CCRF-CEM (IC_50_ = 4.5 µg/mL) and A549 (IC_50_ = 6.25 µg/mL) cell lines. A similar effect has been observed for primary H7PX glioma cell lines [12]. In a recent study, Garcia et al. reported a promising cytotoxic activity of Deroy against leukemia (HL-60, MOLT-3) and lung cancer (A549, NCIH460) cell lines [24]. On the other hand, Deroy does not appear to be cytotoxic against MDA-MB-435 human melanoma or HCT-8 human colon cancer cell lines, at a concentration of 25 µg/mL [25]. In turn, the presented results denote a weak cytotoxic effect (IC_50_ = 25 µg/mL) of Diroy for both tested cell lines (A549 and CCRF-CEM). These results are consistent with our earlier studies, which showed that Diroy also had a weak cytotoxic effect against primary H7PX glioma cells [12]. Moreover, Diroy displayed moderate to weak cytotoxic activities against HL-60 (IC_50_ = 4.5 µM) and K562 (IC_50_ = 36 µM) leukemia cell lines [26].

Apoptosis in cancer cells seems to be the ideal therapeutic aim. This process can be activated by mitochondrial pathways or death receptor pathways. Mitochondria play a key role in regulating apoptosis. The release of mitochondrial cytochrome *c* through *Bax/Bcl-2* activation might activate the effector machinery engaged by p53 to mediate teratogen-induced apoptotic pathways. Some *Bcl-2* family proteins possess pro- and anti-apoptotic properties and are known to control the mitochondrial apoptotic pathway [27,28,29,30,31,32]. This study also evaluated the level of a few genes (*Bax, Bcl-2, Cas-3,* and *TP53*) involved in cancer cell apoptosis. Our findings indicate that all tested compounds displayed higher expression levels of *Bax, Cas-3,* and *TP53* in both cancer cell lines (A549 and CCRF-CEM) when compared to the control cells. Additionally, a significant decrease in mRNA levels of *Bcl-2* was observed in cancer cells, which suggests that the loss of cell viability is connected to apoptosis. The same effect was observed for Roy, Deroy, Diroy, and Parv in primary the H7PX glioma cell line [12]. Burmistrova et al. [10] demonstrated that Parv downregulated the expression of anti-apoptotic *Bcl-2* but did not modulate the expression and/or processing of *Bax* after four hours of treatment in HL-60 cells. Additionally, concentration-dependent processing of initiator (Cas-9 and Cas-8) and executioner (Cas-3) caspases was also observed. Garcia et al. [24] revealed increased levels of Cas-9 and Cas-3/7, but no change in Cas-8 level, in human leukemia MOLT-3 treated with Deroy.

Apoptosis is characterized by the depolarization of mitochondrial membrane potential [28]. The permeability of mitochondrial membrane potential can be altered by an increase in ROS level [32]. Therefore, the present study also evaluated apoptosis parameters, such as ROS levels, the permeability of membrane potential, mtDNA copy number, and DNA damage. In the present study, Roy and Parv demonstrated the strongest cytotoxic activity; these compounds significantly decreased the mitochondrial membrane potential in CCRF-CEM. In turn, in the A549 cell line, Parv caused an increase in mitochondrial membrane potential but Roy did not. Burmistrova et al. reported that Parv treatment in HL-60 human leukemia cells mostly blocked both the generation of intracellular peroxide production and cell death, which suggests that ROS play a key role in Parv cytotoxicity [10]. Additionally, our findings indicate that treatment with Roy induced alterations in mtDNA copy number in the CCRF-CEM leukemia cell line and Parv in the A549 lung cancer cell line. However, while treatment with Roy was associated with a higher mtDNA copy number compared to control untreated cells, Parv decreased the mtDNA copy number. To our knowledge, this is the first report to demonstrate changes in mtDNA copy number in cancer cells after treatment with these compounds. Although, previous studies have noted increased and decreased mtDNA copy numbers in various cancer cell lines, it remains unknown how the changes in mtDNA copy numbers can be responsible for the apoptosis process [33,34]. The present study also evaluated DNA damage in leukemia (CCRF-CEM) and lung (A549) cancer cell lines after treatment with tested abietane diterpenoids. Our findings indicate that Roy and Parv had a genotoxic effect in both cell lines. 

Our results demonstrate, for the first time, that Roy is responsible for the increase in nDNA damage in the *HPRT1* region of A549 cells, while Parv produced an increase in mtDNA damage in both regions (*ND1* and *ND5*) of CCRF-CEM cells.

## 4. Materials and Methods

### 4.1. Plant Material

*P. madagascariensis* (Pers.) Benth. and *P. ecklonii* Benth. (Lamiaceae family) were cultivated in Instituto Superior de Agronomia Campus (Lisbon University), from seeds provided by the Herbarium of the Botanical Garden of Lisbon, Portugal. Their growth was monitored at Parque Botânico da Tapada da Ajuda from cuttings provided by Kirstenbosch National Botanical Gardens, South Africa, collected between 2007 and 2008, in June and September, and voucher specimens were deposited at Herbarium “João de Carvalho e Vasconcellos” of the “Instituto Superior de Agronomia,” Lisboa (LISI), Portugal. 

### 4.2. Isolation of Abietane Diterpenes

The abietane diterpene Deroy was isolated from the essential oil, obtained through the hydrodistillation of *P. madagascariensis* leaves and stems in a Clevenger apparatus, according to Garcia et al. [24]. Diroy and Roy were isolated from acetonic extracts of the aerial parts of the same plant, adapted from the procedure described by Matias et al. [13]. Parv was also isolated from an acetonic extract of *P. ecklonii*, according to Burmistrova et al. [10]. All compounds were then purified by dry-column flash-chromatography, using silica as a stationary phase and a gradient of eluents of increasing polarity (*n*-Hexane:Ethyl acetate) as a mobile phase, according to the same authors [10,13,24]. Their structures were elucidated through NMR spectra assignments and confirmed with authentic samples. NMR spectra were recorded on a Bruker Fourier 300 spectrometer. ^1^H-NMR and ^13^C-NMR spectra were recorded at 300 and 75 MHz, respectively. The chemical shifts (δ) are indicated in ppm, related to TMS, and the coupling constants (J) are indicated in Hz.

**Deroy:** Orange-red needles, ^1^H NMR (CDCl_3_, 300 MHz, ppm): δ 7.34 (1H, s, OH-12), 6.81 (1H, dd, *J* = 9.7, 3.2 Hz, H-7), 6.46 (1H, dd, *J* = 9.7, 3.2 Hz, H-6), 3.16 (1H, hept, *J* = 7.1 Hz, H-15), 2.88 (1H, dt, *J* = 13.3 Hz, H-1β), 2.13 (1H, t, *J* = 3.2 Hz, H-5α), 1.63-1.60 (1H, t, H-2α), 1.52-1.50 (1H, t, H-2β), 1.47-1.46 (1H, m, H-3β), 1.42 (1H, d, *J* = 4.2 Hz, H-1α), 1.23 (1H, s, H-3α), 1.22 (Me-16, overlapped signal), 1.20 (Me-17, overlapped signal), 1.03 (3H, s, Me-20), 1.01(3H, s, Me-18), 0,98 (3H, s, Me-19). ^13^C NMR (CDCl_3_, 75 MHz, ppm): δ; 186.20 (C-14), 183.58 (C-11), 151.34 (C-12), 140.64 (C-9), 139.8 (C-6), 138.6 (C-8), 122.72 (C-13), 121.33 (C-7), 52.23 (C-5), 40.64 (C-4),39.38 (C-3), 35.28 (C-1), 33.40 (C-10), 32.74 (C-19), 24.21 (C-18), 22.94 (C-15), 20.14 (C-17), 19.95 (C-16), 18.81 (C-2), 15.31 (C-20).

**Diroy:** Yellow needles, ^1^H NMR (CDCl_3_, 400 MHz, ppm): δ 7.27 (1H, s, OH-12), 7.25 (1H, s, OH-6β), 4.51 (1H, dd, *J =* 3.3 Hz, *J*7β,6α = 2.0 Hz, H-7β), 4.45 (1H, dd, *J*6α,5α = 4.0 Hz, *J*6α,7β = 2.0 Hz, H-6α), 3.16 (1H, sept, *J*15,16(17) = 7.1 Hz, H-15), 2.93 (1H, d, *J*OH,7β *=* 3.3 Hz, OH-7α), 2.59 (1H, dddd, *J*1β,1α = 12.8 Hz, *J*1β,2α = 3.5 Hz, *J*1β,2β = 3.5 Hz, *J*1β,3β(W) = 1.3 Hz, H-1β), 1.83 (1H, ddddd, *J*2β,1α = 13.4 Hz, *J*2β,1β = 3.5 Hz, *J*2β,2α = 13.9 Hz, *J*2β,3α = 13.4 Hz, *J*2β,3β = 3.4 Hz, H-2β), 1.60 (3H, s, Me-20), ~1.56 (1H, *J*2α,1α = 3.8 Hz, *J*2α,1β = 3.5 Hz, *J*2α,2β = 13.9 Hz, *J*2α,3α = 4.1 Hz, *J*2α,3β = 3.4 Hz, H-2α, overlapped signal), 1.47 (1H, dddd, *J*3,2α = 3.4 Hz, *J*3β,2β = 3.4 Hz, *J*3β,3α = 13.4 Hz, *J*3β(W),1β = 1.3 Hz, H-3β), 1.40 (1H, d, *J*5α,6α = 4.0 Hz, H-5α), 1.25 (3H, s, Me-19), 1.22 (1H, ddd, *J*3α,2α = 4.1 Hz, *J*3α,2β = 13.4 Hz, *J*3α,3β = 13.4 Hz, H-3α), 1.22 (3H, d, J16(17),15 = 7.1 Hz, Me-16), 1.21 (3H, d, *J*_17,15_ = 7.1 Hz, Me-17), 1.18 (1H, ddd, *J*_1__α__,1β_ = 12.8 Hz, *J*_1__α__,2α_ = 3.8 Hz, *J*_1__α__,2β_ = 13.4 Hz, H-1α), 1.04 (3H, s, Me-18). ^13^C NMR (100 MHz, CDCl_3_, ppm): δ 38.5 (C-1, t); 19.0 (C-2, t); 42.3 (C-3, t); 33.8 (C-4, s); 49.5 (C-5, d); 69.4 (C-6, d); 69.2 (C-7, d); 141.0 (C-8, s); 147.5 (C-9, s); 38.6 (C-10, s); 183.5 (C-11, s); 151.1 (C-12, s); 124.3 (C-13, s); 189.1 (C-14, s); 24.3 (C-15, d); 19.9# (C-16, q); 19.8# (C-17, q); 33.5 (C-18, q); 24.0 (C-19, q); 21.6 (C-20, q).

**Roy:** Yellow quadrangular plates, ^1^H-NMR (CDCl_3_, 300 MHz, ppm): δ 7.22 (1H, s, 12-OH), 5.66 (1H, dd, *J* = 2.2, 0.7 Hz, H-7β), 4.31 (1H, s, H-6α), 3.16 (1H, sept, *J* = 7.1 Hz, H-15), 2.63 (1H, d, *J* = 12.8 Hz, H-1β), 2.04 (3H, s, Me-7α-OAc), 1.89– 1.78 (1H, m, H-2β), 1.61 (3H, s, Me-20), 1.55 – 1.46 (2H, m, H-2α and H-3β, overlapped signal), 1.33 (1H, s, H-5α), 1.23 (3H, s, Me-19, overlapped signal), 1.22 (3H, s, Me-17, overlapped signal), 1.21 (1H, s, H-3α, overlapped signal), 1.20 (3H, s, Me-16, overlapped signal), 1.18 (1H, s, H-1α, overlapped signal), 0.94 (3H, s, Me-18). ^13^C-NMR (CDCl_3_, 75 MHz, ppm): δ 185.91 (C11), 183.40 (C14), 169.83 (7α-COCH3), 151.04 (C12), 150.04 (C9), 137.19 (C8), 124.76 (C13), 68.86 (C7), 67.06 (C6), 49.86 (C5), 42.39 (C3), 38.75 (C10), 38.55 (C1), 33.80 (C18), 24.28 (C15), 23.94 (C19), 21.60 (C20), 21.08 (7α-COCH3), 19.97 (C16), 19.84 (C17), 19.10 (C2).

**ParvD:** Orange powder, ^1^H-NMR (CDCl_3_, 300 MHz, ppm): δ 7.93 (2H, d, H-2′ and H-6′), 6.96 (1H, d, *J* = 6.8 Hz, H-14), 6.88 (2H, d, H-3′ and H-5), 6.79 (1H, d, *J* = 6.9 Hz, H-7), 6.41 (1H, s, *J* = 12.5 Hz, H-6), 5.59 (1H, tt, *J* = 4.4 Hz, H-2β), 3.76 (1H, ddd, *J* = 11.4 Hz, H-1β), 3.15 (1H, m, H-15), 2.15 (1H, ddd, *J* = 4.4 Hz, H-3β), 1.74 (1H, dd, *J* = 13.0 Hz, H-1α), 1.64 (3H, s, Me-20), 1.56 (1H, dd, *J* = 11.4 Hz, H-3α), 1.42 (3H, s, Me-19), 1.29 (3H, s, Me-18), 1.18 (3H, d, *J* = 0.8 Hz, Me-16), 1.16 (3H, d, *J* = 2.4 Hz, Me-17). ^13^C-NMR (CDCl_3_, 75 MHz, ppm): δ 178.24 (C12), 166.18 (C7′), 164.84 (C5), 160.58 (C4′), 146.50 (C11), 141.61 (C13), 139.30 (C7), 133.57 (C14), 131.89 (C2′ and C6′), 127.45 (C8), 127.17 (C9), 122.43 (C1′), 118.69 (C6), 115.23 (C3′ and C5′), 67.87 (C2), 45.06 (C3), 43.91 (C10), 38.58 (C4), 38.37 (C1), 33.03 (C18), 30.58 (C19), 26.52 (C15), 25.52 (C20), 21.84 (C16), 21.63 (C17).

### 4.3. Cells and Culture Conditions

Human lung adenocarcinoma A549 (CCL-185; ATCC) cell line and human T lymphoblast CCRF-CEM (CCL-119; ATCC) cell line, used in the experiments, were obtained from American Type Culture Collection (ATCC™, Manassas, VA, USA). Both cell lines were cultured in a humidified incubator at 37 °C and 5% CO_2_. A549 cells were grown in DMEM medium supplemented with 100 units of potassium penicillin and 100 µg of streptomycin sulfate per 1 mL of culture media and 10% (*v*/*v*) heat-inactivated fetal bovine serum (FBS). CCRF-CEM cells were maintained in RMPI 1640 medium and supplemented in the same way as A549. Cell culture reagents were obtained from Lonza (Basel, Switzerland).

### 4.4. MTT Cell Viability Assay

Cell viability was determined by the MTT assay after treatment with increasing concentrations of Roy, Diroy, Deroy, and Parv after 24 h. In brief, A549 cells (1 × 10^4^ cells/well) and CCRF-CEM cells (1 × 10^5^ cells/well) were seeded in a 96-well plate and kept overnight in the incubator (37 °C, 5% CO_2_) for attachment. The next day, the medium was replaced with fresh medium with various concentrations of Roy, Diroy, Deroy, and Parv (0–100 µg/mL). After 24 h of incubation, cells were washed once by centrifugation (300 g for 5 min at 22 °C) and incubated with 0.5 mg/mL of 3-(4,5-dimethylthiazol-2-yl)-2,5-diphenyl tetrazolium bromide (MTT) at 37 °C for 4 h. Then, the MTT was discarded carefully and dimethyl sulfoxide (DMSO) was added to solubilize the formazan crystals. Finally, the absorbance was measured for each well at a wavelength of 570 nm with background subtraction at 630 nm using a Bio-Tek Synergy HT Microplate Reader (Bio-Tek Instruments, Winooski, VT, USA). All experiments were performed in triplicate and the relative cell viability (%) was expressed as a percentage relative to the untreated (control) cells, which was defined as 100%.

### 4.5. Measurement of Intracellular Reactive Oxygen Species (ROS) Level

The relative level of intracellular ROS after treatment with Roy, Diroy, Deroy, and Parv for 1, 2, 12, 24, and 48 h was measured using the redox-sensitive fluorescent dye 2′,7′-dichlorodihydrofluorescein diacetate (DCFH-DA, Molecular probes, Life technologies, Grand Island, NY, USA). Oxidation of DCFH occurs almost exclusively in the cytosol and generates a fluorescent response that is proportional to the intracellular ROS levels. Cells were seeded into black 96-well tissue culture plates with a transparent bottom (Greiner) at a density of 1 × 10^4^ cells/well (A549) or 1 × 10^5^ cells/well (CCRF-CEM) in 50 µL culture medium and cultured overnight in a CO_2_ incubator at 37 °C. The next day, 5 μM of DCFH-DA was added to the cells and incubated further for 45 min at 37 °C in a 5% CO_2_ atmosphere (protected from light). Thereafter, the cells were centrifuged (300× *g* for 5 min at 22 °C) and washed with Hanks’ Balanced Salt Solution (HBSS). Subsequently, cells were incubated with Roy, Diroy, Deroy, and Parv in a CO_2_ incubator at 37 °C, and fluorescence was measured at a 480 nm excitation wavelength and an emission wavelength of 510 nm, using a Bio-Tek Synergy HT Microplate Reader after 1, 2, 12, 24, and 48 h.

### 4.6. Mitochondrial Membrane Potential (MMP)

The MMP was estimated using fluorescent probe JC-1 (5′,6,6′-tetrachloro-1,1′,3,3′-tetraethylbenzimidazolylcarbocyanine iodide). Cells were seeded into black 96-well tissue culture plates with a transparent bottom (Greiner) at a density of 1 × 10^4^ cells/well (A549) or 1 × 10^5^ cells/well (CCRF-CEM) in a 50 µL culture medium and allowed to adhere overnight, and then treated with a IC_50_ concentration of Roy, Diroy, Deroy, and Parv D for 24 h. Finally, the cells were preincubated with 5 μM JC-1 in the HBSS in a CO_2_ incubator at 37 °C for 30 min. Prior to measurements, the cells were centrifuged (300× *g* for 10 min at 22 °C) and then washed twice with the HBSS. The fluorescence was measured on a Bio-Tek Synergy HT Microplate Reader (Bio-Tek Instruments, Winooski, VT, USA) with the filter pairs of 530/590 nm and 485/538 nm. Results are shown as a ratio of fluorescence measured at 530/590 nm to that measured at 485/538 nm (aggregates to monomer fluorescence). All compounds were tested in six duplicates.

### 4.7. Isolation of Total Genomic DNA from Cell Lines

Nuclear and mitochondrial DNA were isolated from 2 × 10^6^ cells by using the commercially available QIAamp DNA Mini Kit, according to the manufacturer’s protocol for extracting DNA from cell lines. DNA concentrations were determined by spectrophotometric measurement of the absorbance at 260 nm and the purities were calculated by a A260/A280 ratio using the Bio-Tek Synergy HT Microplate Reader (Bio-Tek Instruments, Winooski, VT, USA). The purified DNA was stored at −20 °C until further analysis.

### 4.8. Mitochondrial Copy Number

Quantitative real-time PCR was assessed to the relative number of copies of human mitochondrial DNA (mtDNA) using nuclear DNA (nDNA) content as a standard. Two genes were selected for the experiment: *ND1* and *ND5* as a mitochondrial target and *SLCO2B1* and *SERPINA1* genes as a nuclear target. Briefly, real-time PCR amplification was performed in a total volume of 10 µL containing 1 × RT PCR Mix SYBR A (A&A Biotechnology, Gdynia, Poland), 250 nM of each primer, and 1 μL of each analyzed DNA sample. The concentration of each analyzed DNA sample was always 5 ng/μL. The RT-PCR profile consisted of an initial denaturation step for 3 min at 95 °C, 40 cycles at 95 °C for 15 s, 65 °C for 30 s, and 15 s at 72 °C (read fluorescence). Data were collected using the CFX-96 detection system (Bio-Rad, Hercules, CA, USA). Further methodology has been described in our earlier studies [35,36].

### 4.9. Semi-Long-Run qRT-PCR (SLR-qRT-PCR)—Mitochondrial and Nuclear DNA Damage

To quantify mitochondrial DNA (mtDNA) and nuclear DNA (nDNA) damage, a semi-long run quantitative RT-PCR (SLR-qRT-PCR) was performed as described previously, with some modifications [37]. To measure the levels of DNA lesions in the tested region of the mitochondrial or nuclear genome, two fragments of different lengths, i.e., long and small fragments, located in the same mitochondrial/nuclear genomic region were used. The cycle threshold (Ct) values were calculated automatically and the analysis was performed using CFX Manager ^TM^ Software (version 3.1). The cycling condition was as follows: Initial denaturation of 3 min at 95 °C was followed by up to 40 cycles of 15 s at 95 °C, 30 s at 65 °C, and 15 s at 72 °C (short amplicons) or 45 s at 72°C (long amplicons). The SLR-qRT-PCR reaction mix consisted of 1 × RT PCR Mix SYBR A (A&A Biotechnology, Gdynia, Poland), 250 nM of each primer, and 1 ng of template DNA in a total volume of 10 µL per sample. DNA damage was calculated as lesions per 10 kb DNA of each region by including the size of the respective long fragment according to our previous studies [35,36,38].

### 4.10. Gene Expression

Total RNA was extracted from 6 × 10^6^ cells by using ISOLATE II RNA Mini Kit, according to the manufacturer’s instructions. RNA purity and concentration were determined by comparing the absorbances at 260 and 280 nm. The purified RNA samples were stored in TE buffer at −20 °C until further analysis. cDNA was synthesized from total RNA using the High-Capacity cDNA Reverse Transcription Kit. A sample of 1 ng total RNA was used as a template in a total volume of 10 µL, following the manufacturer’s instructions. Next, gene expression was analyzed by TaqMan probe-based real-time PCR assay. TaqMan probes (Life Technologies, Carlsbad, CA, USA) were used to analyze 4 genes (*Bax*, *Bcl-2, Cas-3,* and *TP53*), and *18S RNA* (Life Technologies) was included as the reference gene. qRT-PCR was performed using TaqMan^®^ Real-Time PCR Master Mix (Life Technologies) and Agilent Technologies Stratagene Mx300SP working on MxPro software. The thermal cycling conditions were as follows: 10 min of polymerase activation at 95 °C, followed by 40 cycles of 30 s denaturation at 95 °C and 60 s annealing/extension at 60 °C. Each sample was run in triplicate. The basal expression level was calculated using the Ct method [39].

### 4.11. Statistical Analysis

All analyses were performed using GraphPad Prism^®^ 5.0 Software for Windows (GraphPad Software, Inc., La Jolla, CA, USA). Significant differences between treatment groups were analyzed via one-way ANOVA followed by Dunnett’s post-hoc test. Differences between groups were considered to be significant at a *p* value of <0.05. Statistical significance was defined as a *p* value, * *p* < 0.05, ** *p* < 0.01, *** or *p* < 0.001.

## 5. Conclusions

Our results show that the abietane diterpenoids Roy, Deroy, Diroy, and Parv, isolated from *Plectranthus* spp., possess cytotoxic effects against the CCRF-CEM human leukemia and the A549 lung adenocarcinoma cell lines. All tested compounds induced apoptosis by altering the level of pro- and anti-apoptotic genes. Of the tested diterpenoids, Roy and Parv demonstrated the highest activity in CCRF-CEM and A549 cancer cell lines, through induction of changes in the permeability of mitochondrial membrane potential and ROS levels, and possibly inducing mtDNA or nDNA damage. Our findings show that these abietane diterpenoids may be used as potential natural chemotherapeutic agents in the future; however, in vivo studies are necessary to more precisely determine their clinical possibilities.

## Figures and Tables

**Figure 1 pharmaceuticals-13-00123-f001:**
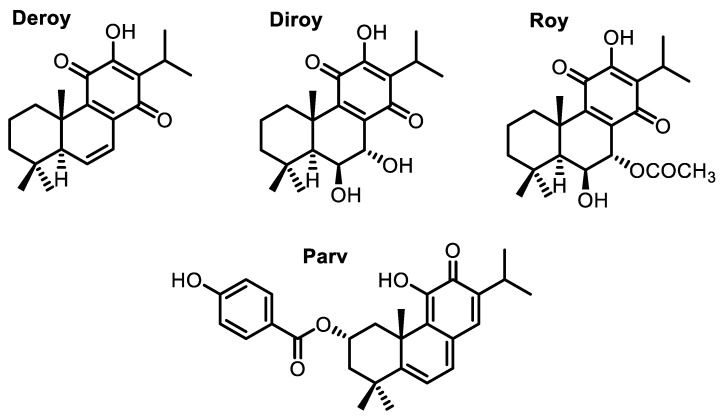
Abietane diterpenes isolated from *P. madagacariensis* and *P. ecklonii.*

**Figure 2 pharmaceuticals-13-00123-f002:**
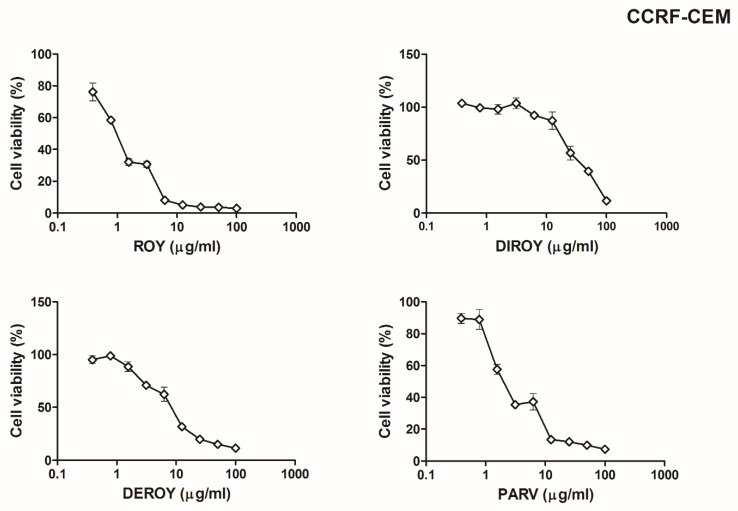
Cell viability response of CCRF-CEM cells using the 3-(4,5-dimethylthiazol-2-yl)-2,5-diphenyl tetrazolium bromide (MTT) assay for 7α-acetoxy-6β-hydroxyroyleanone (Roy), 6,7-dehydroroyleanone (Deroy), 7β,6β-dihydroxyroyleanone6 (Diroy), and Parvifloron D (Parv). CCRF-CEM cells were treated with various concentrations of these compounds (0–100 µg/mL) for 24 h. Values represent the means ± SD as a percent (%) of control.

**Figure 3 pharmaceuticals-13-00123-f003:**
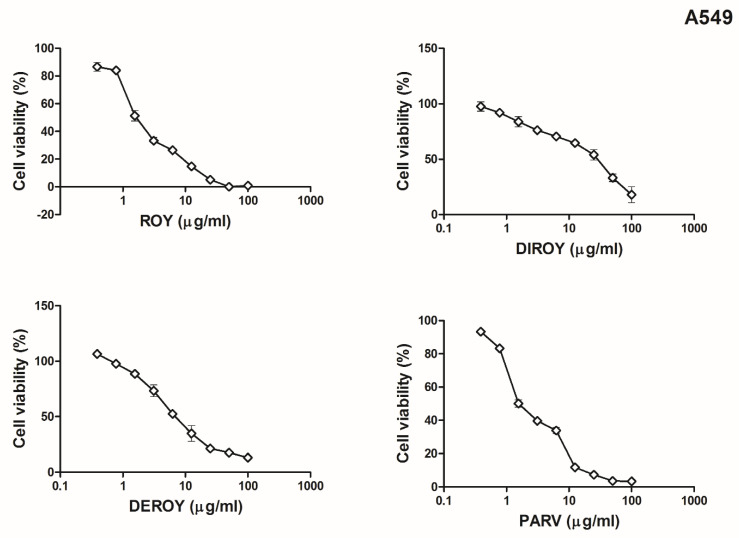
Cell viability response of A549 cells using the MTT assay for Roy, Deroy, Diroy, and Parv. A549 cells were treated with various concentrations of these compounds (0–100 µg/mL) for 24 h. Values represent the means ± SD as a percent (%) of control.

**Figure 4 pharmaceuticals-13-00123-f004:**
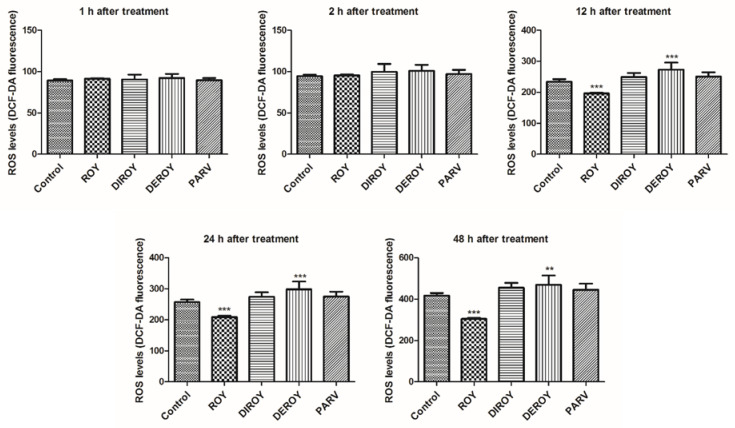
Effect of Roy (IC_50_ = 3 µg/mL), Deroy (IC_50_ = 6.25 µg/mL), Diroy (IC_50_ = 25 µg/mL), and Parv (IC_50_ = 1.15 µg/mL) on intracellular reactive oxygen species (ROS) production. The fluorescence intensity of 2′,7′- dichlorofluorescein (DCF) in A549 cells represented the production of intracellular ROS. Values are expressed as means ± SD. ** *p* < 0.01, *** *p* < 0.001 cells treated with tested compounds vs. control (untreated) cells.

**Figure 5 pharmaceuticals-13-00123-f005:**
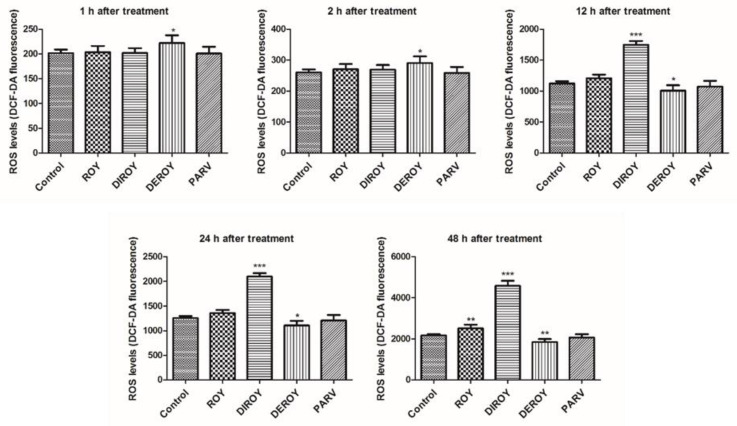
Effect of Roy (IC_50_ = 0.7 µg/mL), Deroy (IC_50_ = 4.5 µg/mL), Diroy (IC_50_ = 25 µg/mL), and Parv (IC_50_ = 1.15 µg/mL) on intracellular reactive oxygen species (ROS) production. The fluorescence intensity of DCF in CCRF-CEM cells represented the production of intracellular ROS. Data are presented as means ± SD. * *p* < 0.05, ** *p* < 0.01, *** *p* < 0.001 cells treated with tested compounds vs. control (untreated) cells.

**Figure 6 pharmaceuticals-13-00123-f006:**
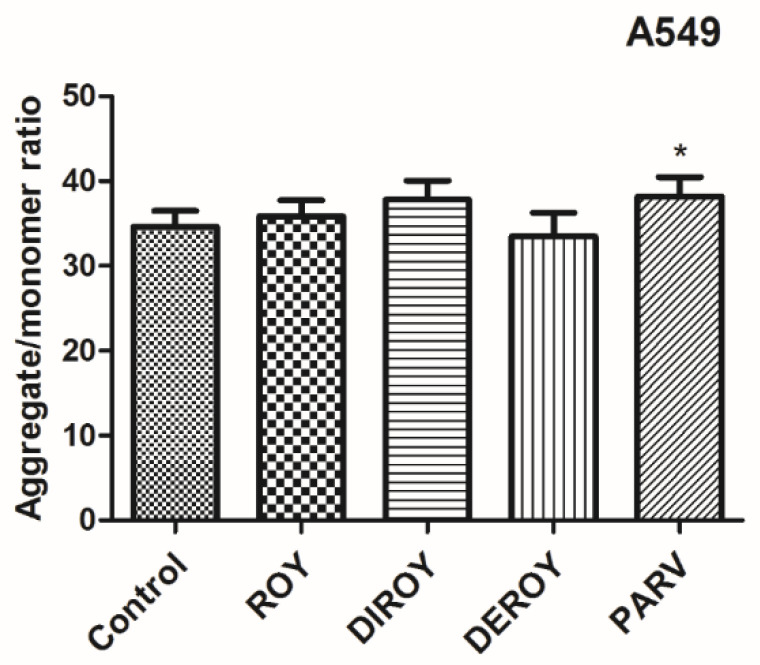
Mitochondrial membrane potential (MMP) is expressed as a ratio of aggregates to monomer fluorescence as quantified with a fluorescent plate reader after 5′,6,6′-tetrachloro-1,1′,3,3′-tetraethylbenzimidazolylcarbocyanine iodide (JC-1) staining. The cells were exposed to Roy (3 µg/mL), Deroy (6.25 µg/mL), Diroy (25 µg/mL), and Parv (1.15 µg/mL) for 24 h. Values are means ± SD. * *p* < 0.05 cells treated with tested compounds vs. control (untreated) cells.

**Figure 7 pharmaceuticals-13-00123-f007:**
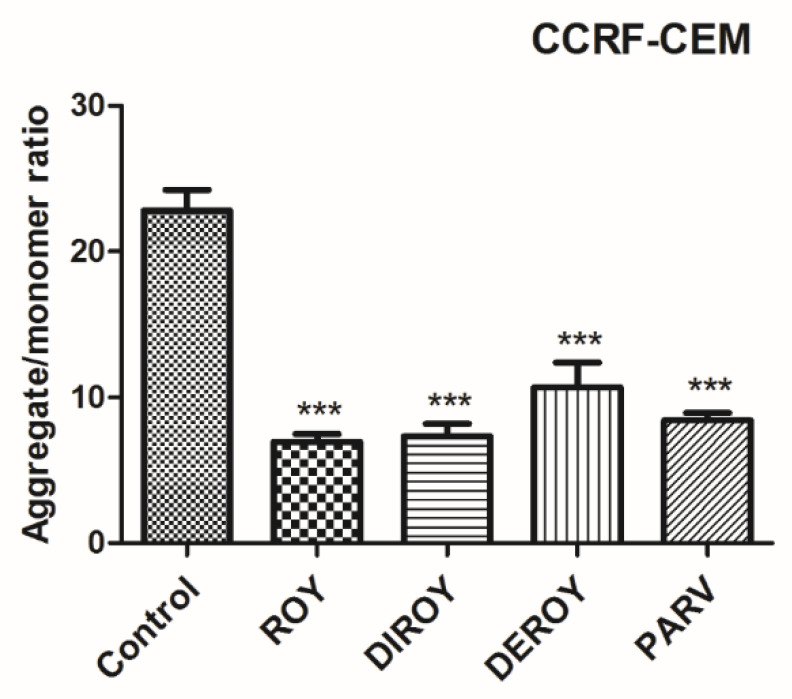
MMP is expressed as a ratio of aggregates to monomer fluorescence as quantified with a fluorescent plate reader after JC-1 staining. The cells were exposed to Roy (0.7 µg/mL), Deroy (4.5 µg/mL), Diroy (25 µg/mL), and Parv (1.15 µg/mL) for 24 h. Values are means ± SD. *** *p* < 0.001 cells treated with tested compounds vs. control (untreated) cells.

**Figure 8 pharmaceuticals-13-00123-f008:**
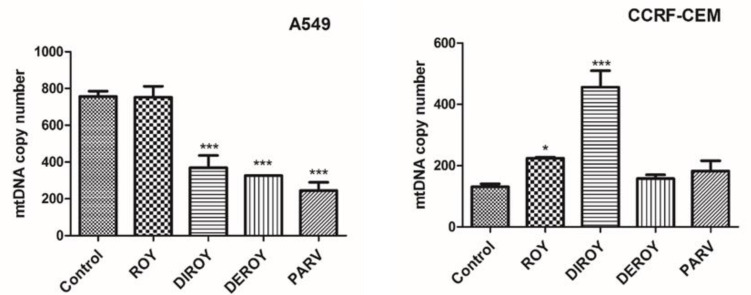
Relative mtDNA copy numbers in A549 and CCRF-CEM cells were detected by real-time PCR. The cells were exposed to Roy (3 µg/mL), Deroy (6.25 µg/mL), Diroy (25 µg/mL), and Parv (1.15 µg/mL) in A549 cells and Roy (0.7 µg/mL), Deroy (4.5 µg/mL), Diroy (25 µg/mL), and Parv (1.15 µg/mL) in CCRF-CEM cells for 24 h. Values are expressed as means ± SD. * *p* < 0.05; *** *p* < 0.001 cells treated with tested compounds vs. control (untreated) cells.

**Figure 9 pharmaceuticals-13-00123-f009:**
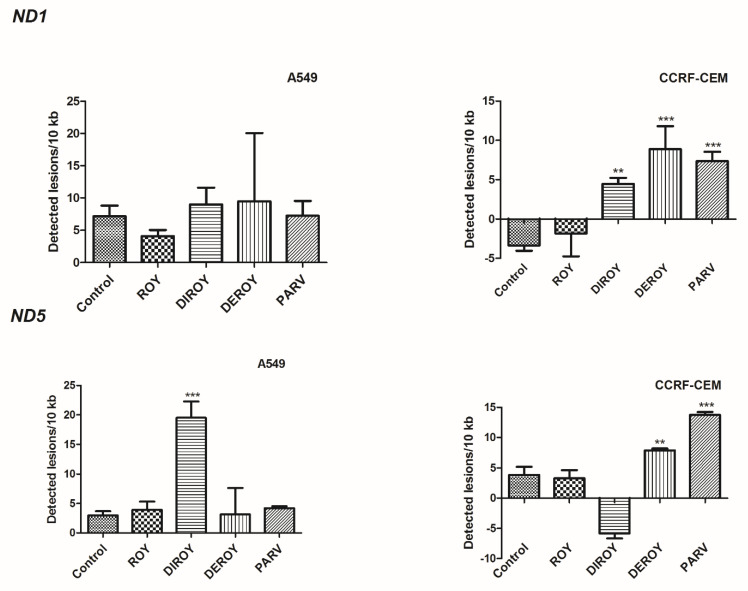
Quantification of mitochondrial DNA (mtDNA) damage per 10 kb of mitochondrial genome measured by SLR-quantitative RT (qRT)-PCR amplification in A549 and CCRF-CEM cells exposed to Roy (3 µg/mL), Deroy (6.25 µg/mL), Diroy (25 µg/mL), and Parv (1.15 µg/mL), and Roy (0.7 µg/mL), Deroy (4.5 µg/mL), Diroy (25 µg/mL), and Parv (1.15 µg/mL), respectively, for 24 h. Data represent the means ± SD of three replicates. ** *p* < 0.01, *** *p* < 0.001 cells treated with tested compounds vs. control (untreated) cells.

**Figure 10 pharmaceuticals-13-00123-f010:**
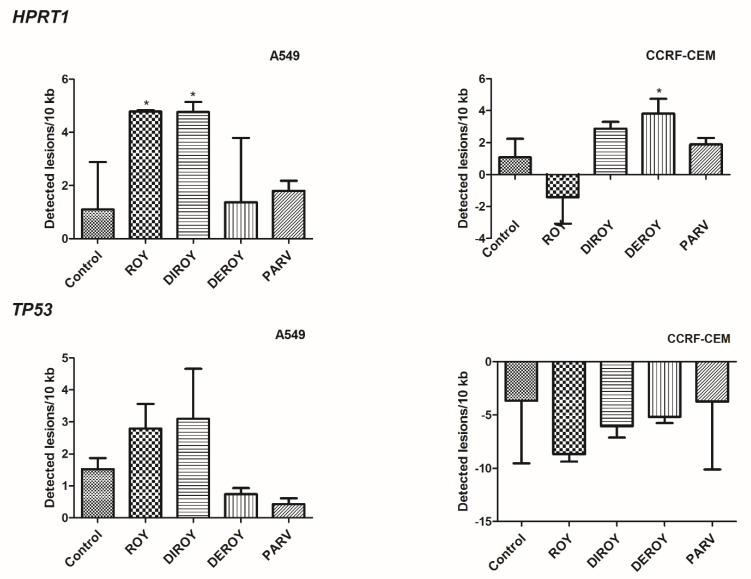
Quantitative measurement of nDNA damage per 10 kb of nuclear genome measured by SLR-qRT-PCR amplification of total DNA from A549 and CCRF-CEM cells exposed to Roy (3 µg/mL), Deroy (6.25 µg/mL), Diroy (25 µg/mL), and Parv (1.15 µg/mL), and Roy (0.7 µg/mL), Deroy (4.5 µg/mL), Diroy (25 µg/mL), and Parv (1.15 µg/mL), respectively, for 24 h. Data represent the means ± SD of three replicates. * *p* < 0.05 cells treated with tested compounds vs. control (untreated) cells.

**Figure 11 pharmaceuticals-13-00123-f011:**
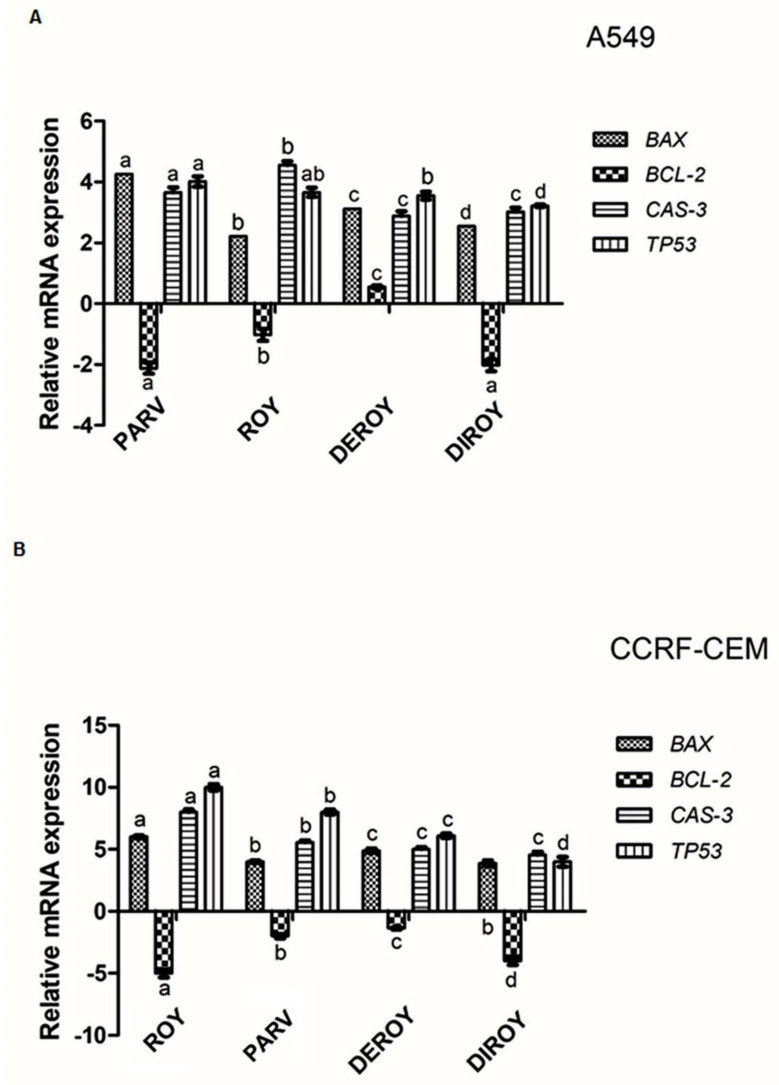
(**A**) Gene expression (*Bax, Bcl-2*, *Cas-3*, and *TP53*) in A549 and (**B**) CCRF-CEM cell lines after 24 h treatment with all the compounds: Roy (3 µg/mL), Deroy (6.25 µg/mL), Diroy (25 µg/mL), and Parv (1.15 µg/mL) in A549 cells and Roy (0.7 µg/mL), Deroy (4.5 µg/mL), Diroy (25 µg/mL), and Parv (1.15 µg/mL) in CCRF-CEM cells. The transcript level of each gene was normalized to the expression of a reference gene (*18S RNA*). Data are presented as a fold change in cells treated with tested compounds vs. untreated, control cells, in which expression levels of the genes were set as 1. The mean values ± SD were calculated in triplicates. The same letter (a, b, c and d) at the same genes is not significantly different at the level of *p* > 0.05.

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
