# Peer review of "Diterpenoids from Plectranthus spp. as Potential Chemotherapeutic Agents via Apoptosis"

_pharmaceuticals, 2020, doi:10.3390/ph13060123_

Round 1
Reviewer 1 Report
ÅšliwiÅ„ski et al., describe “Diterpenoids from Plectranthus spp. as potential chemotherapeutic agents via apoptosis”. The results suggest that the tested diterpenoids, Roy and Parv demonstrated the strongest activity in both human cancer cell lines, changing the permeability mitochondrial membrane potential and ROS levels, and possibly inducing mtDNA or nDNA damage.
Comments:
- Include a better rationalization for choice of leukemia cell line (CCRF-CEM) and lung adenocarcinoma cell line (A549).
- Lines 68: leukaemia cell line?
- Lines 216: A549?
- The current version of the manuscript lacks a strong discussion. The manuscript should be improved.
- The authors should mention on the physiological relevance of the experimental concentrations of Diterpenoids used in the study and whether such concentrations could become clinically achievable.
Author Response
Reviewer #1: Comments to the Authors
ÅšliwiÅ„ski et al., describe “Diterpenoids from Plectranthus spp. as potential chemotherapeutic agents via apoptosis”. The results suggest that the tested diterpenoids, Roy and Parv demonstrated the strongest activity in both human cancer cell lines, changing the permeability mitochondrial membrane potential and ROS levels, and possibly inducing mtDNA or nDNA damage.
Comment 1: Include a better rationalization for choice of leukemia cell line (CCRF-CEM) and lung adenocarcinoma cell line (A549).
Authors: We thank the reviewer for the suggestion.
Thank you very much for reviewer’s suggestion. The cell lines selected by us for research are associated with two cancers which according to WHO occupy top places in statistics on the morbidity and mortality of people around the world. In addition, cancer is the second leading cause of death in the world, accounting for approximately 9.6 million deaths. According to the reviewer’s suggestion this information has been added to manuscript and marked in yellow.
Comment 2: Lines 68: leukaemia cell line?
Authors: We thank the reviewer for the correction. The cell line was exchanged for acute lymphoblast, the correct one.
Comment 3: Lines 216: A549?
Authors: We thank the reviewer for the correction. The cell line was exchanged for CCRF-CEM, the correct one.
Comment 4: The current version of the manuscript lacks a strong discussion. The manuscript should be improved.
Authors: Thank you very much for your comment. We believe the discussion was written based on current literature. There are few papers on the compounds we analysed, which is why we tried to discuss the results from current literature. Nevertheless, according to the reviewer's suggestion, the discussion was rearranged and marked in yellow.
Comment 5: The authors should mention on the physiological relevance of the experimental concentrations of Diterpenoids used in the study and whether such concentrations could become clinically achievable.
Authors: Thank you very much for reviewer’s suggestion. These are preliminary studies on in vitro cell lines and the cytotoxic effect was determined experimentally at different concentrations. On normal cells we did not show a cytotoxic effect in the concentrations presented on the cancer cell lines. The next step in our research will be the in vivo tests and determination of the physiological concentrations of the tested diterpenes.
Reviewer 2 Report
The present manuscript describes the in vitro cytotoxicity assessment of four abietan diterpenoids and at the same time offers an insight regarding possible mechanisms of action (mitochondrial targeted, genotoxic etc.) correlated with the exhibited cytotoxic.
Some changes/revisions are needed regarding the current manuscript.
Firstly regarding, the isolation of the 4 compounds from the plant product, the method is not described in detail. If the method is the same as in references [10], [13] and [20] or is been modified, this aspect should be mentioned in the text. In the same place, the authors mention that the purification of the products was performed by dry flash column chromatography. Here as well, the technique is not described or if the technique is performed according to a previously published method, it must be mentioned. The authors also mention that compound identity validation was achieved by NMR, but details about the used equipment, method or spectra are missing. This data should be included in the manuscript.
In Figure 9 there appears to be a symbol in the upper right corner above the axis that can be fully seen.
In Discussions, the IC50 values ​​of the four compounds are expressed in ug / ml and in some places are compared with other IC50 values mentioned in other studies, which are given in µM. I suggest that the authors make a conversion of values ​​into µM, at least when comparing with other values expressed in the same way, for a clearer text understanding.
Line 340-341, the information within the phrase “On the other hand, Deroy does not appear to be cytotoxic against MDA-MB-435 341 human melanoma or HCT-8 human colon cancer cell lines, at a concentration of 25 µg / mL [27 ] ” is not found in the article corresponding to reference 27. Another source should be given, in case a confusion occurred. Please clarify this aspect.
Line 385, “The abietane diterpene 6β-benzoyloxy-12-O-benzoylroyleanone (Roy-Bz), obtained through semi-synthesis of Roy, is reported in the literature as a PKC-δ selective activator.RoyBz is not genotoxic in human colon cancer cell line HCT-116 because it does not induce DNA damage, nor cause an increase in H2A.X phosphorylation, which is known to be an indicator of double-strand breaks in DNA [37]. ” - I do not understand in the context presented how this information helps considering that Roy, according to the authors, is genotoxic and his derivative is not. Please clarify.
Language must be slightly improved, regarding misspelling errors and sentence topics.
Author Response
RESPONSE TO REVIEWERS’ COMMENTS
Manuscript ID: pharmaceuticals-817757
Title: "Diterpenoids from Plectranthus spp. as potential chemotherapeutic agents via apoptosis”
Author(s): Tomasz ÅšliwiÅ„ski, PrzemysÅ‚aw Sitarek, Ewa SkaÅ‚a, Vera M. S. Isca, Ewelina Synowiec, Tomasz Kowalczyk, MichaÅ‚ Bijak and Patrícia Rijo
Dear Editor,
We appreciate the reviewers’ comments, which have helped us to improve the manuscript. We have carefully considered the suggestions, addressing and incorporating them in the manuscript as detailed below. The modifications in the corrected paper are highlighted in a yellow background.
Reviewer #2: Comments to the Authors
The present manuscript describes the in vitro cytotoxicity assessment of four abietane diterpenoids and at the same time offers an insight regarding possible mechanisms of action (mitochondrial targeted, genotoxic etc.) correlated with the exhibited cytotoxic.
Some changes/revisions are needed regarding the current manuscript.
Comment 1: Firstly regarding, the isolation of the 4 compounds from the plant product, the method is not described in detail. If the method is the same as in references [10], [13] and [20] or is been modified, this aspect should be mentioned in the text. In the same place, the authors mention that the purification of the products was performed by dry flash column chromatography. Here as well, the technique is not described or if the technique is performed according to a previously published method, it must be mentioned. The authors also mention that compound identity validation was achieved by NMR, but details about the used equipment, method or spectra are missing. This data should be included in the manuscript.
Authors: We thank the reviewer for the suggestion. The details of extraction and isolation were added. Additionally, NMR spectra assignments and equipment specificity were also provided.
Comment 2: In Figure 9 there appears to be a symbol in the upper right corner above the axis that can be fully seen.
Authors: We thank the reviewer for the correction. The letter A) and B) in Figure 11 were cut and consequently, it was not possible to read them. They were corrected, accordingly in the right corner and middle right of the Figure 11. The number of the Figure was also correct from 9 to 11.
Comment 3: In Discussions, the IC50 values of the four compounds are expressed in ug / ml and in some places are compared with other IC50 values mentioned in other studies, which are given in µM. I suggest that the authors make a conversion of values into µM, at least when comparing with other values expressed in the same way, for a clearer text understanding.
Authors: Thank you very much for your suggestion. Different biological properties of the tested compounds were determined in different laboratories which in their routine work express the results in different units. Due to the current epidemiological situation in the world and the related difficulties with access to many laboratories where we store our data, we would prefer not to make such a correction at this time. Of course, your suggestion will be implemented in the next manuscript we are preparing.
Comment 4: Line 340-341, the information within the phrase “On the other hand, Deroy does not appear to be cytotoxic against MDA-MB-435 341 human melanoma or HCT-8 human colon cancer cell lines, at a concentration of 25 µg / mL [27 ] ” is not found in the article corresponding to reference 27. Another source should be given, in case a confusion occurred. Please clarify this aspect.
Authors: Thank you very much for reviewer’s comment. We agree it was a mistake and we correct this references in discussion.
Comment 5: Line 385, “The abietane diterpene 6β-benzoyloxy-12-O-benzoylroyleanone (Roy-Bz), obtained through semi-synthesis of Roy, is reported in the literature as a PKC-δ selective activator.RoyBz is not genotoxic in human colon cancer cell line HCT-116 because it does not induce DNA damage, nor cause an increase in H2A.X phosphorylation, which is known to be an indicator of double-strand breaks in DNA [37]. ” - I do not understand in the context presented how this information helps considering that Roy, according to the authors, is genotoxic and his derivative is not. Please clarify.
Authors: Thank you very much for your suggestion. We agree with reviewer’s comment. We added this compound due to its origin and structure to the isolate from the plant. We wanted to give an example of another application of this compound. Nevertheless, the reviewer is right that his insertion is somewhat problematic, so this fragment has been removed from the discussion.
Comment 6: Language must be slightly improved, regarding misspelling errors and sentence topics.
Authors: The manuscript was improved and corrected accordingly.
Round 2
Reviewer 1 Report
I recommend to publish in this present form
Author Response
We thank the reviewer for the comments that help us to improve the manuscript.